# Does Conformation Affect the Analytical Response? A Structural and Infrared Spectral Evaluation of Phenethylamines (2C-H, 25H-NBOH, and 25I-NBOMe) Using In Silico Methodology

**Lívia Salviano Mariotto** [1,2], **Caio Henrique Pinke Rodrigues** [1,2] **and Aline Thais Bruni** [1,2,*]

1 Department of Chemistry, Faculty of Philosophy, Sciences and Letters of Ribeirão Preto, University of São Paulo, Ribeirão Preto 14040-030, SP, Brazil; livia.mariotto@usp.br (L.S.M.); caio.pinke.rodrigues@usp.br (C.H.P.R.)
2 INCT-Forense, Department of Chemistry, Faculty of Philosophy, Sciences and Letters of Ribeirão Preto, University of São Paulo, Ribeirão Preto 14040-030, SP, Brazil
* Correspondence: aline.bruni@usp.br

**Abstract:** The identification of new psychoactive substances (compounds that mimic the effects of outlawed substances) poses a significant challenge due to their rapid emergence and continuous modifications. This phenomenon results in these molecules escaping legal regulation, allowing them to circumvent legislation. The phenethylamine class has garnered attention because its molecules replicate the effects of LSD and are associated with numerous cases of intoxication. In this study, we focused on three phenethylamines—2C-H, 25H-NBOH, and 25I-NBOMe—with crystallographic structures available in the Cambridge Crystallographic Data Center (CCDC) database. We conducted a systematic conformational analysis and compared the structural information obtained. Subsequently, we compared the spectra derived from this analysis with experimental details from the ENFSI database. Structural comparisons were made based on the RMSDs between the lower energy conformations and experimental crystallographic structures. Additionally, structures obtained from direct optimization were compared. We then simulated the spectra based on the X-ray structures and compared them with those in the experimental database. Interpretation was carried out using heat maps and PCA in Pirouette software. Combining in silico methods with experimental approaches provides a more comprehensive understanding of the characterization process of new psychoactive substances (NPSs).

**Keywords:** NPS; phenethylamines; NBOH; NBOMe; 2C

## 1. Introduction

New psychoactive substances, or NPSs, are compounds known as "legal highs" or "designer drugs", among other commercial names. These substances undergo structural modifications to distinguish themselves from prohibited substances, circumventing legislation and fostering a false sense of security during consumption, preserving their recreational appeal. According to the United Nations Office for Drugs and Crime (UNODC), NPSs are substances of abuse that are not controlled by the 1961 Single Convention on Narcotic Drugs or the 1971 Convention on Psychotropic Substances. The term "new" implies that such substances have recently appeared on the market despite being synthesized and known years before [1–4]. Therefore, the rapidity with which such substances are gaining popularity threatens public health and safety. Firstly, since the effects on the human body are not fully understood, these substances are designed to mimic the effects of already-known narcotics. Secondly, this complicates the development of analytical methods for their identification [5,6]. The majority of NPSs reported to UNODC in 111 countries between 2009 and 2017 can be classified into synthetic cannabinoids, synthetic cathinones, and phenethylamines [4].

Thus, one type of NPS that has been gaining notoriety both nationally and internationally is synthetic phenethylamines. With hallucinogenic characteristics, they exhibit a high affinity for 5-HT2A receptors and often found in blotter papers, mimicking the effects of LSD [7] and standing out structurally because they can survive with different radicals, increasing the possibility of the emergence of new substances [4]. The 2C phenethylamines have a primary amine in their structure separated by two phenyl carbons, which is replaced with two methoxy groups in positions 2 and 5 [8,9]. N-benzyl phenethylamines (NBOMes and NBOHs), or substituted phenethylamines, are characterized by the addition of an N-benzyl group to the structure of 2C phenethylamine, being formed by a substituted ring with methoxy groups in positions 2 and 5 (ring A) and with a substituted methoxy group in position 1 of the second ring (ring B) in the case of NBOMes or hydroxyl in the case of NBOHs [10,11]. These substances represent a serious social problem, mainly linked to records of poisonings and related deaths [11].

When the seizure of unknown substances occurs, it is necessary to prove their nature so that the law is applied correctly. The Scientific Working Group for Analyzing Seized Drugs (SWGDRUG) proposes some recommendations on how identifications should be carried out. These tests are separated into categories A, B, and C, in decreasing order of selectivity. They recommend that for correct identification of any seized substance, a category A test and another from any category, or in the absence of category A tests, two category B tests and one category C test, are carried out [12]. As the appearance of NPSs is very accelerated, there is great difficulty in developing reliable methods for their identification and quantification. Modifications in their molecules occur very quickly, implying the absence of reference standards [13], meaning that such tests are not always accurate enough to identify NPSs, even category A ones.

In order to solve this problem regarding the identification of NPSs, in silico methods (chemometrics [14], quantum chemistry [15], molecular dynamics [16], statistics [17]) can be applied in combination with the experimental methods proposed by SWGDRUG. Such methods can provide important information more quickly and with low investments. Furthermore, there is no need for any type of government authorization for the study of such seized substances, since their structures and spectra are computationally simulated [13]. The methods mentioned can differentiate spectra at points that are not perceptible to the naked eye, are capable of interpreting a large set of data in a simplified way, and, in general, do not require elaborate sample preparation procedures [11].

In this article, 2C phenethylamines, NBOMes, and NBOHs are described due to growing international interest [18].

## 2. Materials and Methods

### 2.1. Step I: Selection of Molecules

The molecules 2C-H, 25H-NBOH, and 25I-NBOMe (Figure 1) were chosen because their respective crystallographic structures are registered in the Cambridge Crystallographic Data Center (CCDC) database. These structures are identified by the codes EKUMOP (2C-H) [19], EKUKUT (25H-NBOH) [19], and (25I-NBOMe) [20]. The selection of these molecules was also based on a preference for simple structures of phenethylamines without substituents. However, at the time of writing this work, the only crystallographic structure of NBOMe found in the database was the one associated with an iodine atom.

**Figure 1.** Representation of the structures obtained from CCDC and used in the study.

## 2.2. Step II: Construction of Inputs

Avogadro software (1.2.0, University of Pittsburgh Department of Chemistry, Pittsburgh, Pennsylvania, United States of America) [21] was used to generate the input file for the ORCA software (5.0.2, Max-Planck-Institut f¨ur Kohlenforschung, Mülheim an der Ruhr, Germany) [22], employing the Becke 3-parameter Lee–Yang–Parr (B3LYP) and mixed Perdew–Burke–Ernzerhof and Hartree–Fock exchange energy (PBE0) functionals (input specifications are provided in the Supplementary Materials). The basis employed was balanced polarized triple-zeta derived from the def2-TZVP bases with minor modifications for elements 5 s, 6 s, 4 d, and 5 d as well as iodine (dhf-TZVP). The B3LYP hybrid functional has a 20% Hartree–Fock exchange, in which the non-localized approach is combined with the energy functional from the generalized gradient approximation, showing good performance when reproducing energy gaps in many materials. It has been previously employed in studies of phenethylamines, as has PBE0, which contains a predefined amount of exact exchange [23–27].

## 2.3. Step III: Determination of Minimum Energy Structures

### 2.3.1. Step III.1. Determination of the Minimum Energy Structure from Crystallographic Structures

The crystallographic structure was added to the program, generating the input file for direct optimization with the two functionals using the ORCA 5.0.2 software [22].

### 2.3.2. Step III.2. Determination of Minimum Energy Structures from Systematic Search

The conformational analysis consists of studies in which the geometries of conformers, and consequently their energies, are evaluated according to the calculation method inserted in the force fields of molecular mechanics. In this case, the experimental values must be as close as possible to the ab initio calculations, making the force field perform better. Conformers are identified by conformational searching, in which an algorithm rotates the geometry of a molecule by repeatedly varying the torsion angles with predetermined values. Minimizing the energy of the force field means that all possible conformations are found, and a potential energy surface can be obtained [28,29]. In this work, a systematic search was carried out, which combines a series of structural parameters according to each conformation [30].

The number of possible conformers (S) for the molecule can be obtained by Equation (1):

$$S = \left(\frac{360}{\theta i}\right) N \tag{1}$$

$\theta i$ = dihedral angle increment;
N = number of angles with free rotation.

Avogadro software was used to perform the systematic conformational analysis, with the MMFF94 force field [31], in accordance with what was found in the literature for molecules similar to those studied. The software searched by rotating the molecule's dihedrals to obtain its conformers and then optimized its geometry to the lowest energy position. With the help of ORCA software, conformers were reoptimized with B3LYP-D3BJ [32,33] and PBE0-D3BJ [34] functionals (input specifications are presented in the Supplementary Materials). The vibrational frequencies obtained in the outputs from the conformers confirm that the structure is at its true minimum and does not present negative values.

The Boltzmann distribution values (Equation (2)) were obtained from the Gibbs free energy shown in the outputs. The distribution shows the probability of finding conformers at a given temperature [29]. The following equation can measure the diversity of conformers in equilibrium at 298.15 K:

$$p_i = \frac{\exp\left(-\frac{E_i}{k_B T}\right)}{\sum_{j=1}^{N} \exp\left(-\frac{E_i}{k_B T}\right)} \tag{2}$$

$k_B$ (Boltzmann constant) = 0.001987;
N = number of conformers;
$E_i$ = electronic energy of conformer i in the ground state.

### 2.4. Step IV: Structure Comparison

The RMSD values were used to compare the structures based on the distance between the conformations obtained after optimization with the crystallographic structures used in the study. The RMSD denotes the contribution of each bond concerning some torsion angle; the more significant the RMSD, the greater the impact on chemical space in subsequent generations of conformers [35]. This calculation is expressed using Equation (3):

$$\text{RMSD} = \frac{1}{N}\sqrt{\sum_{i=1}^{Natoms}\left(ri(t1) - ri(t2)\right)2} \tag{3}$$

### 2.5. Step V: Infrared Spectra

The infrared spectra of the studied molecules were plotted according to the results of the respective calculation outputs. The outputs provide values of theoretical wavelengths and intensities, making it possible to generate graphs and compare them with the infrared spectra available in the European Network of Forensic Science Institutes (ENFSI) database. When theoretical spectra are obtained, corrections are necessary, according to the DFTs, due to the non-treatment of anharmonicity effects in the calculations. These are known as the scaling factor. The values used were found in the literature (1.0044 for B3LYP and 0.9944 for PBE0) and adequately applied [36,37].

One of the comparisons used between the transmittance values obtained was the Pearson correlation (Equation (4)). This approach measures the linear relationship between two numbers of transmittances (corresponding to each molecule compared) present at the same frequency (stipulated from 400 cm$^{-1}$ to 4000 cm$^{-1}$ with an interval of 1 in 1 cm$^{-1}$). The correlation value varies from $-1$ to $+1$. In this case, the value of $+1$ corresponds to a perfect correlation between the values; they are equal. The value $-1$ indicates the inverse correlation between the values, and 0 shows that the values do not correlate [38]. To make this analysis visual, the data were represented in a heat graph.

$$\rho = \frac{\sum_{i=1}^{n}\left(x_i - \bar{x}\right)\left(y_i - \bar{y}\right)}{\sqrt{\sum_{i=1}^{n}\left(x_i - \bar{x}\right)^2}\sqrt{\sum_{i=1}^{n}\left(y_i - \bar{y}\right)^2}} \tag{4}$$

A chemometric method was also used to compare transmittances since these are multivariate data. The method in question was principal component analysis (PCA). In this case, the objective was to condense relevant information from the data set into a smaller

number of variables to reduce the system's dimensions. The measurements are described in matrix form, in which the lines correspond to the samples and the columns correspond to the variables. The operation performed was [39].

X = T. $P^T$ + E;

X = original data matrix (m × n);

P = loading matrix;

T = score matrix;

E = residual matrix.

## 3. Results and Discussion

### 3.1. Step I. Adjustment of Structures

Crystallographic structures obtained from the CCDC structure database of 2C-H, 25H-NBOH, and 25I-NBOMe are represented respectively in Figure 2A–C:

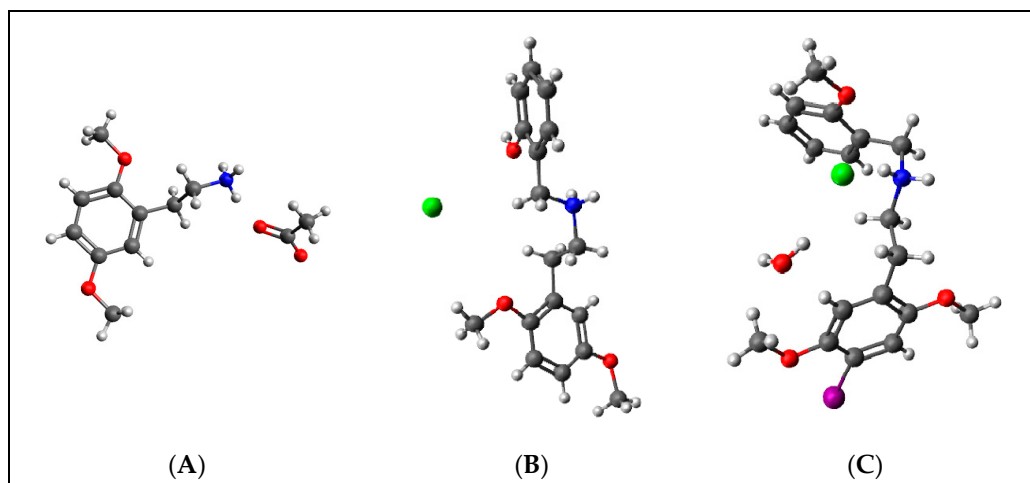

**Figure 2.** Representation of the structures used in the study as they were experimentally crystallographic, being (**A**) 2C-H, (**B**) 25H-NBOH, and (**C**) 25I-NBOMe.

All crystals of the structures depicted in Figure 2 were obtained in their acidic form, with protonated nitrogen. This indicates that the X-ray structures are accompanied by an acid structure that dissociates, with its hydrogen bonding to the amine present in the molecule. Using the Avogadro software, these structures were edited, removing the hydrogen, and the anion from the acid structure was present before the calculations began.

### 3.2. Step II. Construction of Inputs

At this stage, input files were created in the Avogadro software based on each crystallographic structure. These originated from two files referring to the different B3LYP-D3BJ and PBE0-D3BJ functionals, with the respective keywords for each calculation described in Step III.

### 3.3. Step III. Determination of Minimum Energy Structures

3.3.1. Step III.1. Determination of the Minimum Energy Structure from Crystallographic Structures

The six inputs (three molecules, with two functionals each) obtained in Step II were optimized directly from the crystallographic structure. Responses to the values of frequencies and intensities of the infrared spectra of such structures were received. The processing of this data will be addressed in Step V.

3.3.2. Step III.2. Determination of Minimum Energy Structures from Systematic Search

The three crystallographic structures were subjected to a systematic search using Avogadro software. Three conformers were found for the 2C-H structure, twenty-seven

for the 25H-NBOH structure, and twenty-seven for the 25I-NBOMe structure. Each of these conformers was optimized by ORCA software with the B3LYP-D3BJ functional and PBE0-D3BJ. Only Gibbs energy values of the outputs with all positive frequencies (true minimum) for each conformer were used to calculate the Boltzmann distribution. Furthermore, the frequencies and intensities of the infrared spectra were also calculated, and these data will be explored in Step V. The graphs with the Gibbs energies are in Figures 3–8.

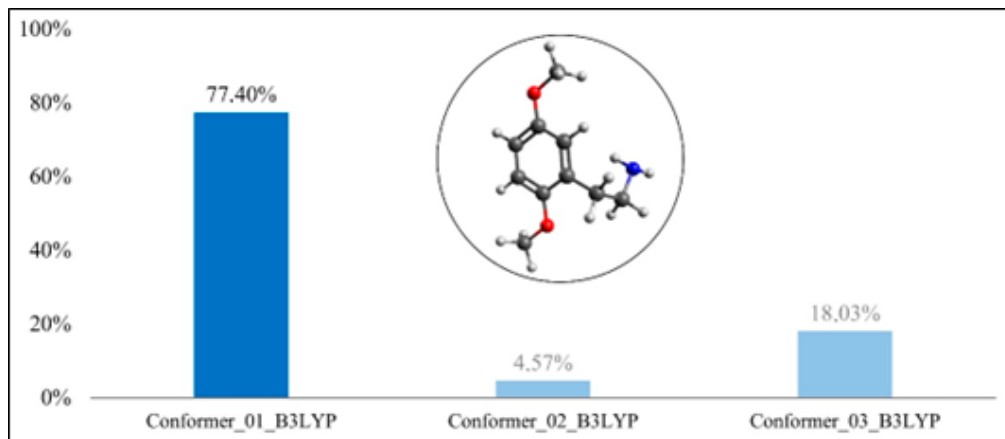

**Figure 3.** Boltzmann distribution of the molecule's conformers for 2C-H(B3LYP-D3BJ).

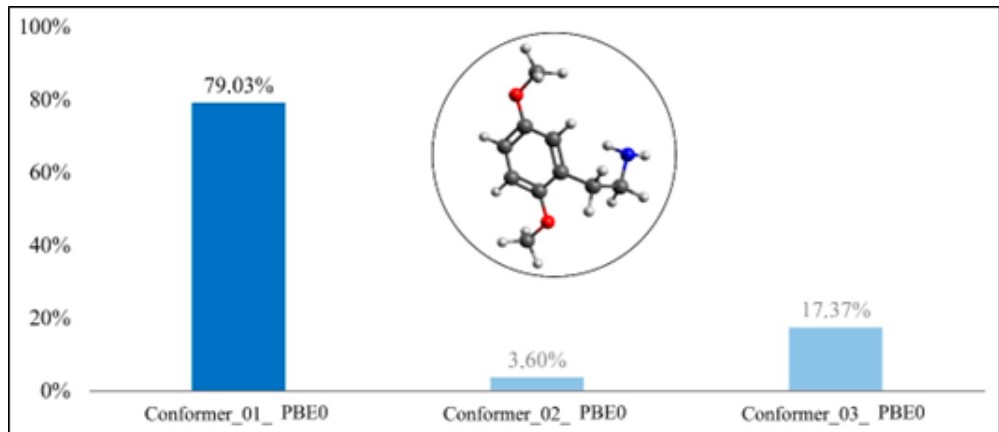

**Figure 4.** Boltzmann distribution of the molecule's conformers for 2C-H(PBE0-D3BJ).

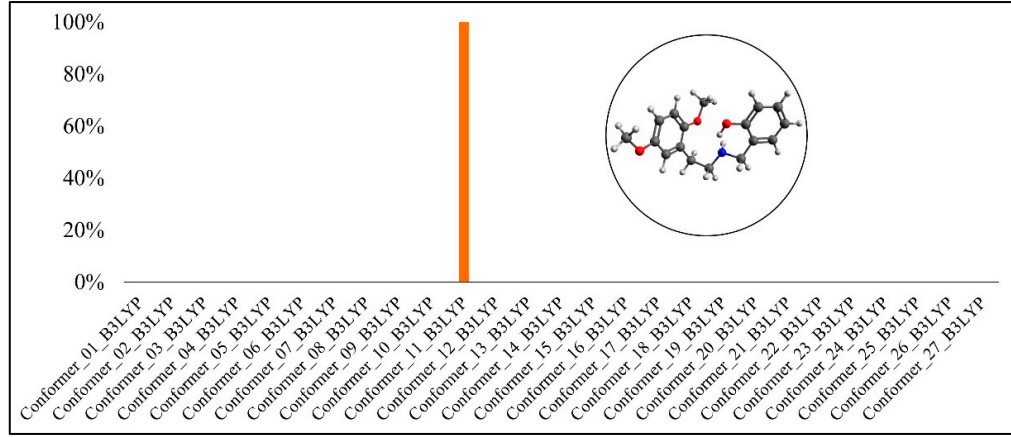

**Figure 5.** Boltzmann distribution of the molecule's conformers for 25H-NBOH(B3LYP-D3BJ).

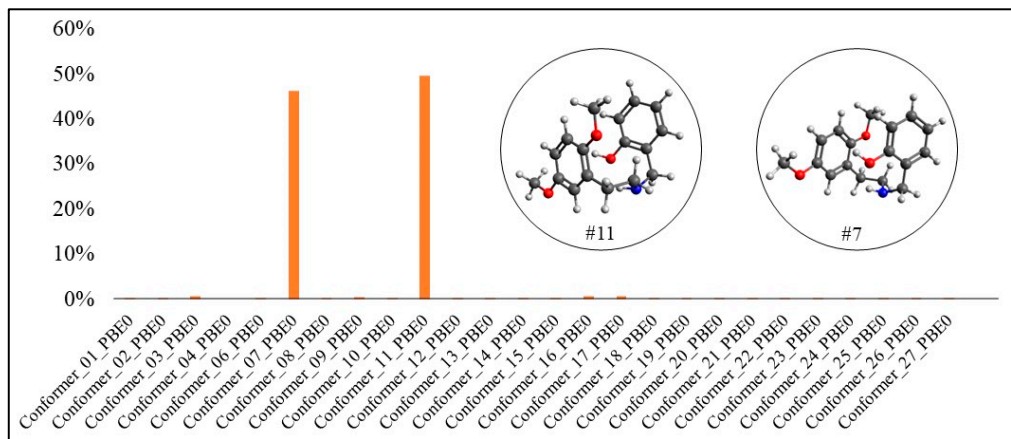

**Figure 6.** Boltzmann distribution of the molecule's conformers for 25H-NBOH(PBE0-D3BJ).

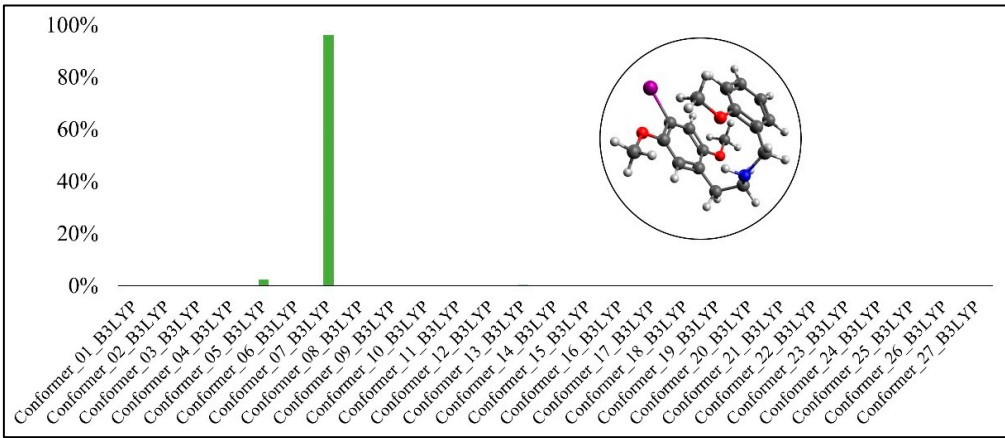

**Figure 7.** Boltzmann distribution of the molecule's conformers for 25I-NBOMe(B3LYP-D3BJ).

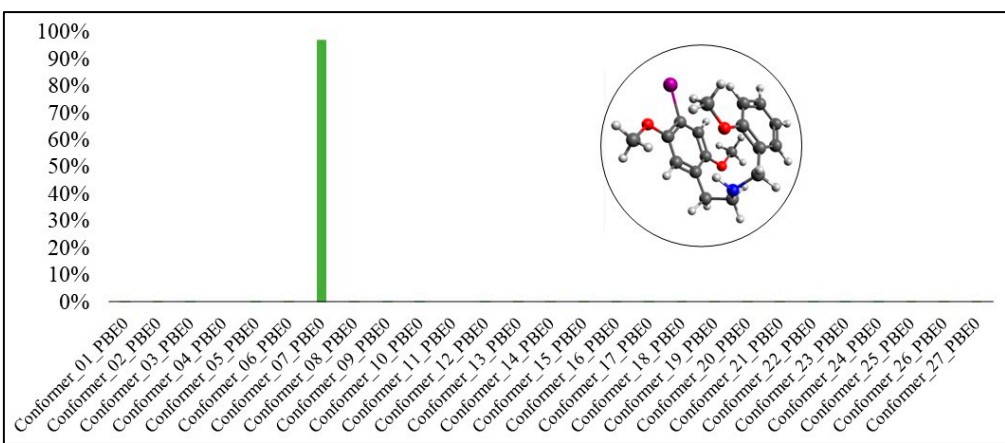

**Figure 8.** Boltzmann distribution of the molecule's conformers for 25I-NBOMe(PBE0-D3BJ).

In Figures 3 and 4, we can observe that the conformer with the largest Boltzmann population of the 2C-H molecule was conformer #1, in both functionals. In the optimization with B3LYP-D3BJ, the Boltzmann population was 77.40%, whereas with PBE0-D3BJ, it was 79.03% for conformer #1.

In Figures 5 and 6, it is possible to observe that conformer #11 presented the highest Boltzmann population in both functionals, with 99.92% for B3LYP-D3BJ and 49.64% for PBE0-D3BJ. However, for PBE0-D3BJ, conformer #7 also had a Boltzmann population very close to that of conformer #11, 46.26%.

In Figures 7 and 8, conformer #7 presented the highest Boltzmann population in both DFTs, with 96.20% for B3LYP-D3BJ and 97.07% for PBE0-D3BJ.

### 3.4. Step IV: Structure Comparison

The direct optimization molecules (Step III.1) and the energy minima from the systematic search (Step III.2) were compared with their respective crystals using RMSD. The closer such values are to 0, the more similar the structures. Table 1 shows the RMSD values of the direct optimization and the energy minimum (conformer 1) of the 2C-H molecule.

**Table 1.** Total enthalpies (Eh).

| Molecule | B3LYP-D3BJ | PBE0-D3BJ |
|---|---|---|
| 2C-H | −594.9350010 | −594.569666 |
| 25H-NBOH | −940.347749 | −939.771682 |
| 25I-NBOMe | −1274.733976 | −1274.146490 |

Regarding RMSD, the optimized structures show excellent structural proximity to the crystal, with the one optimized with B3LYP-D3BJ being the closest. The structures resulting from the systematic search are slightly more different from the crystallographic structures than those optimized directly. However, the values are still 1 angstrom smaller for the 2C-H structure, with B3LYP-D3BJ slightly smaller.

For the 25H-NBOH molecule (Table 2) and 2C-H, the structures resulting from direct optimization presented RMSDs closer to the crystallographic structure than those from conformational analysis.

**Table 2.** Total Enthalpies and Gibbs Free Energy (Eh).

| Molecule | Total Enthalpies | | Gibbs Free Energy | |
|---|---|---|---|---|
| | B3LYP-D3BJ | PBE0-D3BJ | B3LYP-D3BJ | PBE0-D3BJ |
| 2C-H #1 | −594.936933 | −594.571658 | −594.990244 | −594.624771 |
| 25H-NBOH #11 | −940.364575 | −939.782176 | −940.431149 | −939.847965 |
| 25I-NBOMe #7 | −1274.744139 | −1274.155596 | −1274.817674 | −1274.229062 |

In both cases, those that were optimized with B3LYP-D3BJ were closer. However, conformer #7 optimized with PBE0-D3BJ presented an RMSD closer to the original molecule than conformer #11 with PBE0-D3BJ.

Following the same pattern as previous results, the results for the 25I-NBOMe molecule (Table 3) indicated that the directly optimized structures presented more satisfactory RMSD results than those from the minima.

**Table 3.** RMSD values between optimizations and crystals for the 2C-H molecule structure.

| Direct Optimization | | Conformational Analysis Conf. #1 | |
|---|---|---|---|
| B3LYP-D3BJ | PBE0-D3BJ | B3LYP-D3BJ | PBE0-D3BJ |
| 0.037 | 0.047 | 0.582 | 0.588 |

As shown in Table 3, the minimum energy structures optimized with PBE0-D3BJ presented an RMSD value that was more satisfactory than when using B3LYP-D3BJ. A phenomenon in relation to the energy minima found in the 25H-NBOH and 25I-NBOMe molecules is their "closed" conformation, unlike in crystallographic structures as shown in Tables 4 and 5. One explanation for the phenomenon is the π–π interactions in benzene dimers. These interactions are non-covalent and occur between the aromatic rings, providing specific molecular stability, and are essential in the study of drug design [40,41].

**Table 4.** RMSD values between optimizations and crystals for the molecule structure 25H-NBOH.

| Direct Optimization | | Conformational Analysis | | |
|---|---|---|---|---|
| | | Conf. #11 | | Conf. #7 |
| B3LYP-D3BJ | PBE0-D3BJ | B3LYP-D3BJ | PBE0-D3BJ | PBE0-D3BJ |
| 0.165 | 0.207 | 2.079 | 4.013 | 2.670 |

**Table 5.** RMSD values between optimizations and crystals for the structure of the 25I-NBOMe molecule.

| Direct Optimization | | Conformational Analysis Conf. #7 | |
|---|---|---|---|
| B3LYP-D3BJ | PBE0-D3BJ | B3LYP-D3BJ | PBE0-D3BJ |
| 0.343 | 0.373 | 2.303 | 2.290 |

The most common configurations in benzene dimers with such interactions are cofacial sandwich geometry, a slip-stacked or parallel displaced geometry, and T-shaped geometry. To explain the preference for each type of conformation, Hunter and Sanders developed a simple model based on the quadrupole moment of benzene; that is, how the rings position themselves will depend on electrostatic interactions [42–44]. The strength of such interactions also depends on the distance the rings are from each other and their orientation [40,42], meaning that even with the rings' steric restrictions, they can still interact with each other and bring considerable stability to the system [40].

Podeszwa, Bukowski, and Szalewicz present the potential energy surface of a benzene dimer without any substituent through ab initio calculations, concluding that the two minimum isoenergetic points were those conformers with T-shaped and parallel displaced configurations [45]. According to Lee, Kim, Jurecka, Tarakeshwar, Hobza, and Kim, the presence of substituents, regardless of whether they are donors or acceptors, favors the parallel displaced conformation due to their effects on the electrostatic interactions between the rings [46]. The studies agree with the results obtained in the present work, since the minima from the conformational analysis of the 25I-NBOMe molecule present a parallel

displaced conformation, and the minima of 25H-NBOH have conformations close to a T-shaped conformation.

While not the primary focus of the present study, it is important to highlight the potential influence of solvents on the stability of conformers. Solvents can impact stability through electrostatic and dipole–dipole interactions, as well as their polarity and dielectric constant. These solvent molecules interact with the conformers, thereby modifying their stability and conformation through the aforementioned intermolecular forces [47,48].

### 3.5. Step V: Infrared Spectra

With the calculation outputs, it was possible to construct theoretical infrared spectra for each conformation. After obtaining such spectra, they were compared with the experimental spectra present in the ENFSI database. Examples of the graphic comparison is in Figures 9–11. Spectra from different origins were selected for the same molecule, and all were placed on the graph. In this case, the idea was also to contrast the experimental spectra to verify the agreement between them. The comparison was made using a correlation matrix between the transmittances of the spectra, as shown in Figure 12. Values that correspond to 1 mean that the transmittances compared are equal for each wavelength. Green coloring shows correlation values that vary between 1 and 0.3, yellow for values between 0.2 and 0.1, and red from 0 to $-0.2$.

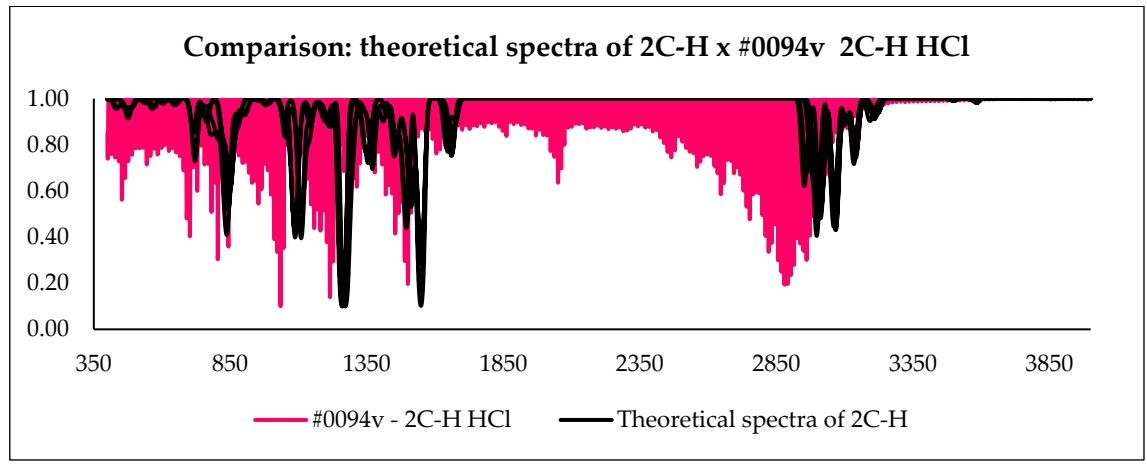

**Figure 9.** Comparison between all theoretical spectra of 2C-H and experimental spectra.

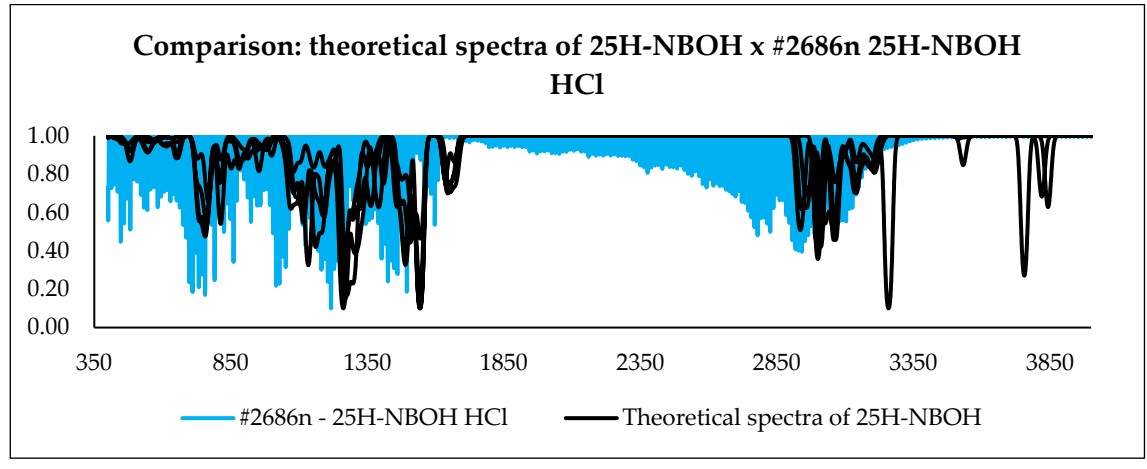

**Figure 10.** Comparison between all theoretical spectra of 25H-NBOH and experimental spectra.

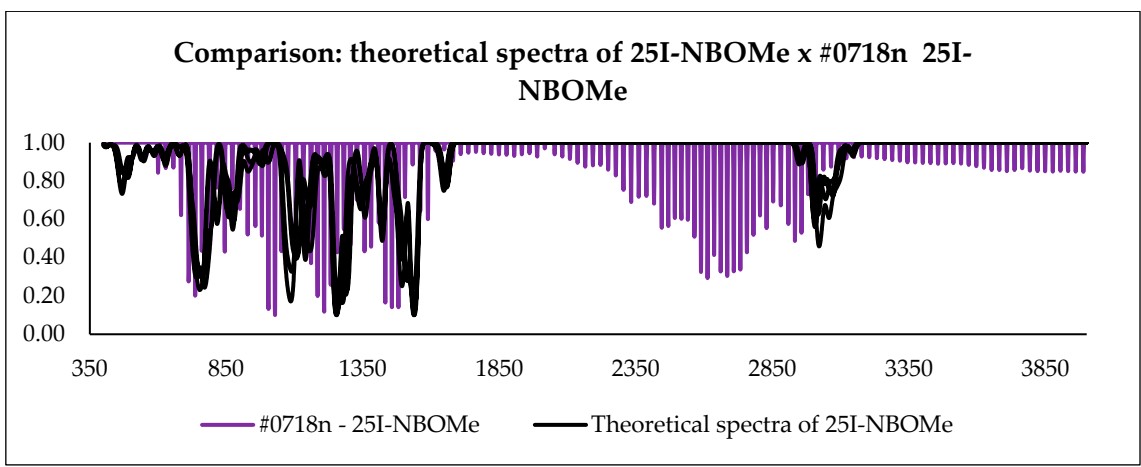

**Figure 11.** Comparison between all theoretical spectra of 25I-NBOMe and experimental spectra.

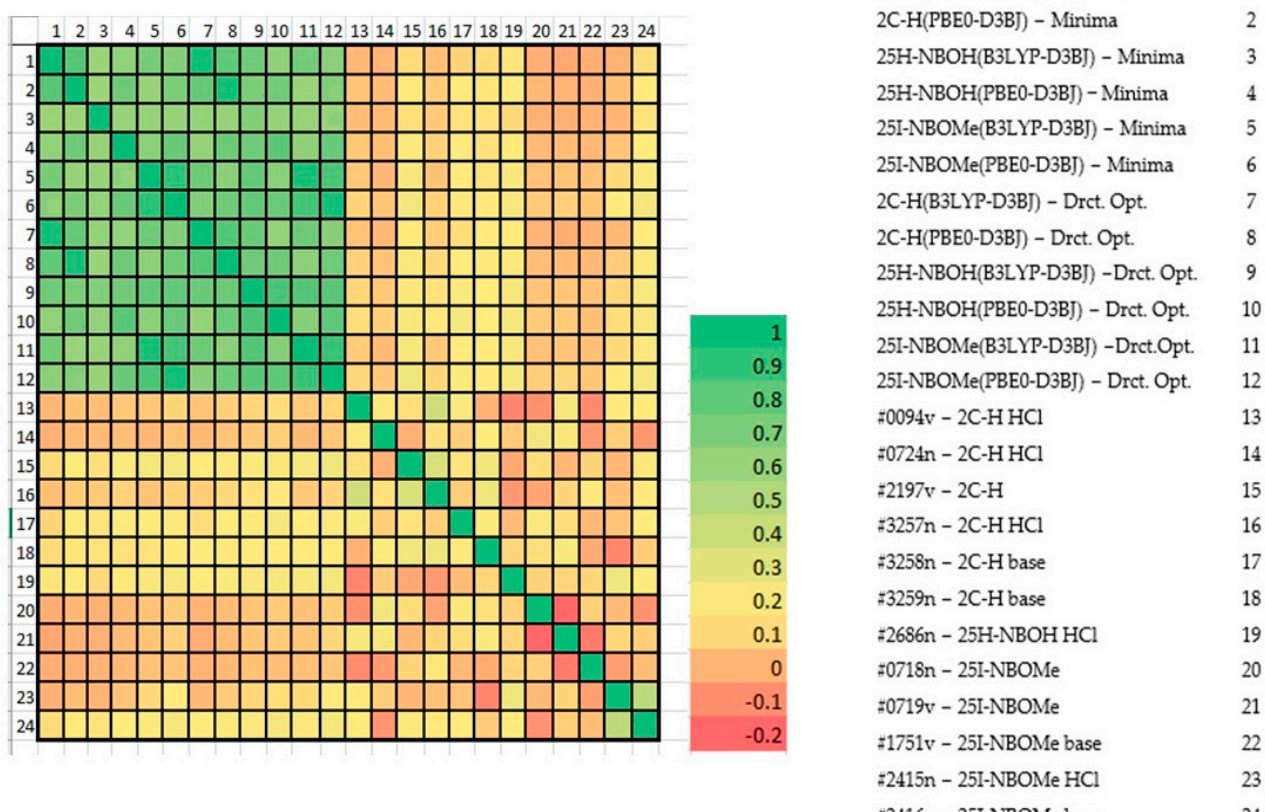

**Figure 12.** Heat plot of theoretical and experimental spectra.

From the graph presented in Figure 12, it is possible to observe significant similarity between the theoretical spectra of the molecules. The experimental spectra do not resemble each other, not even those from the same molecule. The only molecule that presents a certain similarity between its two theoretical spectra was 25I-NBOMe (#2415/#2416). With the same set of theoretical and experimental spectra used previously, a PCA was performed, with the result in Figure 13. In this case, the objective was to verify the spectral behavior for the substances.

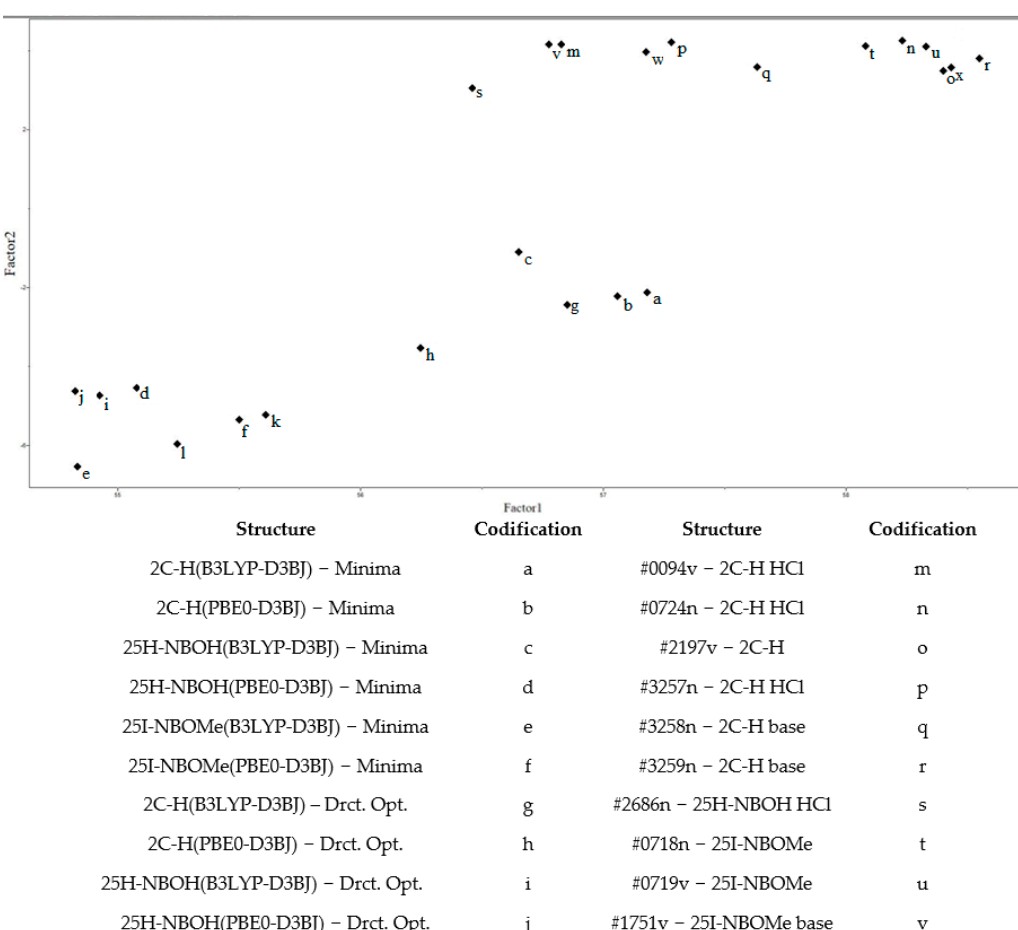

| Structure | Codification | Structure | Codification |
|---|---|---|---|
| 2C-H(B3LYP-D3BJ) – Minima | a | #0094v – 2C-H HCl | m |
| 2C-H(PBE0-D3BJ) – Minima | b | #0724n – 2C-H HCl | n |
| 25H-NBOH(B3LYP-D3BJ) – Minima | c | #2197v – 2C-H | o |
| 25H-NBOH(PBE0-D3BJ) – Minima | d | #3257n – 2C-H HCl | p |
| 25I-NBOMe(B3LYP-D3BJ) – Minima | e | #3258n – 2C-H base | q |
| 25I-NBOMe(PBE0-D3BJ) – Minima | f | #3259n – 2C-H base | r |
| 2C-H(B3LYP-D3BJ) – Drct. Opt. | g | #2686n – 25H-NBOH HCl | s |
| 2C-H(PBE0-D3BJ) – Drct. Opt. | h | #0718n – 25I-NBOMe | t |
| 25H-NBOH(B3LYP-D3BJ) – Drct. Opt. | i | #0719v – 25I-NBOMe | u |
| 25H-NBOH(PBE0-D3BJ) – Drct. Opt. | j | #1751v – 25I-NBOMe base | v |
| 25I-NBOMe(B3LYP-D3BJ) – Drct. Opt. | k | #2415n – 25I-NBOMe HCl | w |
| 25I-NBOMe(PBE0-D3BJ) – Drct. Opt. | l | #2416n – 25I-NBOMe base | x |

**Figure 13.** Principal Component Analysis (theoretical and experimental spectra).

Using PCA and from Figure 10, it was possible to observe the distribution of samples in the new axis system, in this case, Factors 1 and 2. No type of preprocessing was applied. Factor 1 held approximately 98% of the original information used for the PCA. Factor 2 contained only 0.52% of the information, which indicates that only one dimension (Factor 1) was necessary to describe the data.

Analyzing Factor 1 (horizontally), the directly optimized theoretical spectra of the molecules 25H-NBOH(PBE0-D3BJ), 25H-NBOH(B3LYP-D3BJ), 25I-NBOMe(B3LYP-D3BJ), 25I-NBOMe(PBE0-D3BJ), and 2C-H(PBE0-D3BJ) appear close together in a group. However, the minima obtained from conformational analysis of the 25I-NBOMe molecule (B3LYP-D3BJ/PBE0-D3BJ) occur in the same group. A group of five samples formed by the minima of 25H-NBOH(B3LYP-D3BJ), 25H-NBOH(PBE0-D3BJ), 2C-H(B3LYP-D3BJ), and 2C-H(PBE0-D3BJ) and a direct optimization of the 2C-H(B3LYP-D3BJ) molecule are in the central part of the image.

The experimental spectra are in the upper right part of Figure 10, showing that most of the spectra that underwent conformational analysis are closer to the experimental ones, especially those in which the molecules did not have any halogen. Analyzing Factor 2, it was possible to observe a separation between the theoretical and experimental groups. However, the amount of information in this main component is minimal, indicating that differences exist but are minimal compared to the entire data set.

## 4. Conclusions

This work aimed to verify phenethylamines' structural and spectral similarity (analogous to the 2C, NBOH, and NBOMe structures) computationally simulated with their experimental structures. For this, calculations were made with two types of functionals for each molecule. The infrared data survey was carried out for structures optimized directly from crystallographic data and those obtained through conformational analysis. The results showed that infrared depends on structural variation for these molecules.

Structural determination through systematic search showed structures with a specific interaction between their non-covalent aromatic rings. These π–π interactions promote molecular stability and condition the molecule into a conformation with a certain proximity between the rings.

The heat graph comparison shows that the theoretical spectra are very different from the experimental ones, even with several experimental spectra of the same molecule. It is important to note that the experimental spectra in the same database for the same molecule also present discrepant values between them. This indicates that great care must be taken to compare the spectra of suspected substances directly with those obtained from databases. These spectra can have different bands even when dealing with the same molecule, as they may not have the same type of sample preparation and come from different devices, since researchers deposit their results in the database and there is no standard for the development of such spectra. Considering that the experimental spectra do not even resemble each other, even if they are the same molecule, it is very unlikely that they would be similar to the theoretical spectra of such a molecule. Furthermore, theoretical spectra portray bands of a completely pure substance, which experimentally is practically impossible, with bands often appearing due to contamination. PCA showed that the infrared values from the systematic search were closer to those from direct optimization. This indicates that, structurally, the systematic search is essential for surveying the spectroscopic properties of phenethylamines.

**Supplementary Materials:** The following supporting information can be downloaded at: https://www.mdpi.com/article/10.3390/psychoactives3010006/s1, Table S1: 2C-H: crystal conformation, xyz coordinates, and conformers, Table S2: 25H-NBOH: crystal conformation, xyz coordinates, and conformers, Table S3: 25I-NBOMe: crystal conformation, xyz coordinates, and conformers [49].

**Author Contributions:** Conceptualization, L.S.M. and A.T.B.; methodology, L.S.M.; validation, L.S.M., C.H.P.R. and A.T.B.; writing—original draft preparation, L.S.M.; writing—review and editing, L.S.M. and C.H.P.R.; supervision, A.T.B. All authors have read and agreed to the published version of the manuscript.

**Funding:** This research received funding from the Conselho Nacional de Desenvolvimento Científico e Tecnológico (CNPq, process 151152/2022-5), Instituto Nacional de Ciência e Tecnologia Ciências Forenses (INCT Forense/CNPq, project 465450/2014-8; process 465450/2014-8; process 104496/2023-1), Coordenação de Aperfeiçoamento de Pessoal de Nível Superior (CAPES, Financial Code 001), and Programa Nacional de Cooperação Acadêmica—Segurança Pública e Ciências Forenses (PROCAD/CAPES process 16/2020).

**Institutional Review Board Statement:** Not applicable.

**Informed Consent Statement:** Not applicable.

**Data Availability Statement:** The experimental spectra data can be found in the ENFSI database.

**Conflicts of Interest:** The authors declare no conflicts of interest.

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
