# Peer review of "Does Conformation Affect the Analytical Response? A Structural and Infrared Spectral Evaluation of Phenethylamines (2C-H, 25H-NBOH, and 25I-NBOMe) Using In Silico Methodology"

_psychoactives, doi:10.3390/psychoactives3010006_

Round 1
Reviewer 1 Report
Comments and Suggestions for Authors
In their study, Mariotto and colleagues explore the structural and vibrational properties of phenethylamine-based new psychoactive substances. The topic is relevant to the field and holds potential interest for the Psychoactives scientific community. However, the manuscript has significant issues in both style and data analysis that need to be addressed before it can be considered for publication. The paper has language inconsistencies; some figures and text sections are in the authors' native language instead of English, which is not acceptable for an English publication. Additionally, the manuscript fails to provide a detailed and critical discussion of the discrepancies between theoretical and experimental data, which is a crucial aspect of the research. Given these issues, the manuscript is not suitable for publication in its current form. Major revisions are required. The authors need to ensure language consistency throughout the paper and provide a more in-depth analysis of the discrepancies between their findings and existing data. Specific points for revision are detailed below.
Main issues
- According to the authors, 25I-NBOMe has an iodine atom; however, this atom is absent from the structure provided in Figure 1. The authors should clarify this point.
- The authors should justify why they chose B3LYP and PBE0. They also should provide information about the basis set when discussing the construction of inputs in the main text.
- I suggest the authors to specifically mention B3LYP-D3(BJ) and PBE0-D3(BJ), since they considered dispersion corrections. Neglecting this information in the main text could mislead readers that dispersion corrections were not taken into account.
- The SI mentions some references - however, no reference is given in the file. The authors should clarify this.
- The procedure of Adjustment of Structures in Section 3.1 needs to be better explained. I suggest the authors include the XYZ coordinates of the structures computed in the SI.
- Figures 3-4: Some numbers have a comma separation, rather than a dot separation. Please correct.
- The structures of the distinct conformers investigated in this work should be available in the SI.
- Table 1: numerical values in Hartree should be shown with a maximum of six decimal digits.
- The authors should clarify how solvent effects could affect their conformational analysis.
- Figure 10 needs to be improved. The text in the plot is overlapped, leading to major problems in visualising the data. Furthermore, some parts are not written in English.
- The values used for the scaling factors should be explicitly mentioned.
- The authors are analysing the infrared spectrum of NPS substances. However, not a single infrared spectrum is shown in their manuscript. At least one figure comparing the theoretical and experimental IR spectra should be present.
- The main conclusion of this work is that there are significant mismatches between the computed and the experimental data. The specific reasons for such mismatch need to be further investigated in detail.
Minor issues
Abstract: NPS should be defined
Line 51: There is an extra space after 'structure of'
Line 68: Statistics rather than statistic.
Line 103: The font size seems different from other sections. A similar issue can be found in other sections of the manuscript.
Line 107: Sentence 'Conforming conformers are identified by conformational searching' needs to be rewritten.
Line 117: Symbols must be the same as those used in the equation.
Line 131: Equation 2 must be written in English.
Line 170: The section title should be written in English.
Comments on the Quality of English Language
Minor English language errors are present throughout the text. I recommend the authors thoroughly proofread and correct these in the revised version.
Author Response
Reviewer 1
Main issues
We appreciate the reviewer's observation. We have corrected the manuscript according to this recommendation and we hope we have satisfactorily responded to all suggestions.
1) According to the authors, 25I-NBOMe has an iodine atom; however, this atom is absent from the structure provided in Figure 1. The authors should clarify this point.
The molecule has been corrected.
2) The authors should justify why they chose B3LYP and PBE0. They also should provide information about the basis set when discussing the construction of inputs in the main text.
The justification is in the line 95.
3) I suggest the authors to specifically mention B3LYP-D3(BJ) and PBE0-D3(BJ), since they considered dispersion corrections. Neglecting this information in the main text could mislead readers that dispersion corrections were not considered.
Modification accepted.
4) The SI mentions some references - however, no reference is given in the file. The authors should clarify this.
References have been added.
5) The procedure of Adjustment of Structures in Section 3.1 needs to be better explained. I suggest the authors include the XYZ coordinates of the structures computed in the SI.
The explanation has been improved in line 179. Coordinates have been added in SI.
6) Figures 3-4: Some numbers have a comma separation rather than a dot separation. Please correct.
Adjusted.
7) The structures of the distinct conformers investigated in this work should be available in the SI.
Conformers have been added to the SI.
8) Table 1: numerical values in Hartree should be shown with a maximum of six decimal digits.
Adjusted.
9) The authors should clarify how solvent effects could affect their conformational analysis.
Commented on line 279.
10) Figure 10 needs to be improved. The text in the plot is overlapped, leading to major problems in visualizing the data. Furthermore, some parts are not written in English.
Adjusted.
11) The values used for the scaling factors should be explicitly mentioned.
Added in line 151.
12) The authors are analyzing the infrared spectrum of NPS substances. However, not a single infrared spectrum is shown in their manuscript. At least one figure comparing the theoretical and experimental IR spectra should be present.
Spectra have been added to images 9, 10, and 11.
13) The main conclusion of this work is that there are significant mismatches between the computed and the experimental data. The specific reasons for such mismatch need to be further investigated in detail.
Added in line 379.
Minor issues
14) Abstract: NPS should be defined.
Adjusted.
15) Line 51: There is an extra space after 'structure of.'
Adjusted.
16) Line 68: Statistics rather than statistics.
Adjusted.
17) Line 103: The font size seems different from other sections. A similar issue can be found in other sections of the manuscript.
Adjusted.
18) Line 107: Sentence 'Conforming conformers are identified by conformational searching' needs to be rewritten.
Adjusted.
19) Line 117: Symbols must be the same as those used in the equation.
Adjusted.
20) Line 131: Equation 2 must be written in English.
Adjusted.
21) Line 170: The section title should be written in English.
Adjusted.
Reviewer 2 Report
Comments and Suggestions for Authors
Does conformation influence the analytical response? A strucural and spectral evaluation of phenethylamines (2C-H, 25H-3 NBOH and 25I-NBOMe) using in silico methodology.
The purpose of the paper is to demonstrate if in silico methodology can be useful to identify and characterize NPS. The authors selected the crystallographic structures in the CCDC database of three phenethylamines – 2C-H, 25H-NBOH, and 25I-NBOMe.
Minimum Energy Structure from Crystallographic Structures was determined. Moreover, a systematic conformational search was carried out, using Avogadro software, on the three crystallographic structures.
At the end, the authors compared the theoretical infrared spectra for each conformation with the experimental spectra present in the ENFSI database.
Comments:
The manuscript is interesting and presents new and useful contexts. There are no similar publications in the literature on the subject. The conclusions are consistent with the evidence and the references are appropriate and exhaustive. The tables provide detailed information. English language is fine. However, the style is not very well edited and only infrared spectra were considered as analytical data.
Some improvements are recommended as follows:
- Please, add the link, in the manuscript or in the literature reference, where you can download the ORCA software (https://www.faccts.de/orca/#);
- Please explicit the acronym “B3LYP” page 3 line 95;
- Please explicit the acronym “PBE0” page 3 line 95;
- The format of all references should follow the style request of the journal;
- The contribution of each author to the article should be specified;
- Portuguese words are found in the manuscript. They should be replaced with the corresponding English terms (page 5 line 170, page 9 Table 5, page 11 Figure 10);
- The numbering of the tables is incorrect. It is reversed. Table 4 appears before Table 3 (page 8). Renumbering tables correctly;
- A change in the title of the manuscript is requested given that only infrared spectra were considered as analytical data.
The paper can be accepted after minor revision.

Author Response
Reviewer 2
1) Please add the link in the manuscript or in the literature reference, where you can download the ORCA software (https://www.faccts.de/orca/#);
Added in reference number 28.
2) Please explicit the acronym "B3LYP" on page 3, line 95;
Adjusted.
3) Please explicit the acronym "PBE0" on page 3, line 95;
Adjusted.
4) The format of all references should follow the style request of the journal;
Adjusted.
5) The contribution of each author to the article should be specified;
Adjusted.
6) Portuguese words are found in the manuscript. They should be replaced with the corresponding English terms (page 5, line 170, page 9 Table 5, page 11 Figure 10);
Adjusted.
7) The numbering of the tables is incorrect. It is reversed. Table 4 appears before Table 3 (page 8). Renumbering tables correctly;
Adjusted.
8) A change in the title of the manuscript is requested, given that only infrared spectra were considered as analytical data.
Adjusted.
The authors would like to thank the reviewers for all their dedication in helping to improve the quality of the article, contributing immensely to the enhancement of the final work.